# GIST: Greedy Independent Set Thresholding for Max-Min Diversification with Submodular Utility

**Matthew Fahrbach**[*]
Google
fahrbach@google.com

**Srikumar Ramalingam**[*]
Google
rsrikumar@google.com

**Morteza Zadimoghaddam**[*]
Google
zadim@google.com

**Sara Ahmadian**
Google
sahmadian@google.com

**Gui Citovsky**
Google
gcitovsky@google.com

**Giulia DeSalvo**
Google
giuliad@google.com

## Abstract

This work studies a novel subset selection problem called *max-min diversification with monotone submodular utility* (MDMS), which has a wide range of applications in machine learning, e.g., data sampling and feature selection. Given a set of points in a metric space, the goal of MDMS is to maximize $f(S) = g(S) + \lambda \cdot \mathtt{div}(S)$ subject to a cardinality constraint $|S| \leq k$, where $g(S)$ is a monotone submodular function and $\mathtt{div}(S) = \min_{u,v \in S : u \neq v} \mathrm{dist}(u, v)$ is the *max-min diversity* objective. We propose the GIST algorithm, which gives a $1/2$-approximation guarantee for MDMS by approximating a series of maximum independent set problems with a bicriteria greedy algorithm. We also prove that it is NP-hard to approximate within a factor of $0.5584$. Finally, we show in our empirical study that GIST outperforms state-of-the-art benchmarks for a single-shot data sampling task on ImageNet.

## 1 Introduction

Subset selection is a ubiquitous and challenging problem at the intersection of machine learning and combinatorial optimization, finding applications in areas such as feature selection, recommendation systems, and data summarization, including the critical task of designing pretraining sets for large language models [59, 3]. Data sampling in particular is an increasingly important problem given the unprecedented and continuous streams of data collection. For example, one self-driving vehicle generates ~80 terabytes of data daily from LiDAR and imaging devices [34], and academic datasets have scaled dramatically from 1.2M images in ImageNet [54] to 5B in Laion [56].

Subset selection often involves balancing competing objectives: we rely on the *utility* (or weight) of individual items to prioritize them, while simultaneously trying to avoid selecting duplicate or near-duplicate items to ensure *diversity*. When selecting a small subset, this process should guarantee that the chosen set is a good representation of the original dataset—often called *coverage*. The intricate trade-offs between utility, diversity, and coverage are often expressed through an objective function. It is often a significant challenge to design efficient algorithms with strong approximation guarantees for constrained subset selection problems, even when leveraging tools and techniques from combinatorial optimization such as submodular maximization, $k$-center clustering, and convex hull approximations.

This paper studies a novel subset selection problem called *max-min diversification with monotone submodular utility* (MDMS). Given $n$ points $V$ in a metric space, a nonnegative monotone submodular

---

[*]Equal contribution.

39th Conference on Neural Information Processing Systems (NeurIPS 2025).

function $g : 2^V \to \mathbb{R}_{\geq 0}$, diversity strength $\lambda \geq 0$, and cardinality constraint $k$, our goal is to solve

$$S^* = \underset{S \subseteq V}{\arg\max} \ g(S) + \lambda \cdot \texttt{div}(S) \tag{1}$$

$$\text{subject to } |S| \leq k,$$

where $\texttt{div}(S) = \min_{u,v \in S : u \neq v} \text{dist}(u, v)$ is the max-min diversification term [2, 40, 38]. Maximizing the submodular term $g(S)$ aims to select points that are the most valuable or informative. A purely utility-driven algorithm, however, can lead to redundancies where very similar points are chosen. To counteract this, we add the $\lambda \cdot \texttt{div}(S)$ term to encourage selecting points that are well spread out in the metric space, which effectively penalizes subsets with closely clustered elements. Note that $\lambda$ is a knob that controls the trade-off between these two terms, acting as a regularization strength for subset selection problems.

The most relevant prior work is by Borodin et al. [13]. They combine a monotone submodular utility and the *max-sum diversity* term $\sum_{u,v \in S} \text{dist}(u, v)$, in contrast to our max-min $\texttt{div}(S)$ in (1). They give a greedy $1/2$-approximation algorithm and extend their results to general matroid constraints via local search. An earlier contribution by Gollapudi and Sharma [26] combined a *linear utility* (instead of a monotone submodular function) and the max-sum diversity term, subject to the strict constraint $|S| = k$, to design a ranking system for search engines. These approaches focus on maximizing the sum of pairwise distances, but our max-min formulation ensures that the selected points have high utility and are maximally separated, leading to less redundant and more representative subsets.

## 1.1 Our contributions

We summarize the main contributions of this work below:

- In Section 2, we formalize the MDMS problem and present a simple $0.387$-approximation algorithm as a warm-up to show how the competing terms in the objective function interact.

- In Section 3, we present the GIST algorithm, which achieves a $1/2$-approximation guarantee for MDMS by approximating a series of maximum independent set problems with a bicriteria greedy algorithm and returning the best solution. In the special case of *linear utility functions* $g(S) = \sum_{v \in S} w(v)$, we prove that GIST offers a stronger $2/3$-approximation guarantee.

- In Section 4, we prove that it is NP-hard to approximate MDMS to within a factor of $0.5584$ via a careful reduction from the maximum coverage problem. For linear utilities, we *match the guarantees* of GIST and prove a tight $(2/3 + \varepsilon)$-hardness of approximation, for any $\varepsilon > 0$, assuming $P \neq NP$. In the more restricted case of a linear utility function and the Euclidean metric, we prove APX-completeness.

- In Section 5, our experiments show that GIST outperforms baselines for MDMS on synthetic data (in particular the classic greedy algorithm). Then we show that GIST can be used to build better single-shot subsets of training data for an image classification benchmark on ImageNet compared to margin sampling and $k$-center algorithms, demonstrating the benefit of optimizing for a blend of utility and diversity.

## 1.2 Related work

**Submodular maximization.** Submodular maximization subject to a cardinality constraint $k$ is an NP-hard problem, i.e., $\max_{S \subseteq V : |S| \leq k} g(S)$. For monotone submodular functions, Nemhauser et al. [47] proved that the greedy algorithm achieves a $(1 - 1/e)$-approximation, which is asymptotically optimal unless $P = NP$ [23]. The non-monotone case is substantially less understood, but Buchbinder and Feldman [14] recently gave a $0.401$-approximation algorithm and Qi [48] improved the hardness of approximation to $0.478$. Over the last decade, there has also been significant research on distributed algorithms for submodular maximization in the MapReduce [44, 43, 9, 10, 39, 33] and low-adaptivity models [7, 19, 22, 8, 5, 16, 17].

**Diversity maximization.** The related *max-min diversification* problem $\max_{S \subseteq V : |S| = k} \texttt{div}(S)$ has a rich history in operations research due to its connection to facility location, and is also called the *p-dispersion* problem. It uses the *strict constraint* $|S| = k$; otherwise, if $|S| \leq k$ we can maximize $\texttt{div}(S)$ with two diametrical points. Tamir [58] gave a simple greedy $1/2$-approximation algorithm for

max-min diversification, and Ravi et al. [51] proved $(1/2 + \varepsilon)$-inapproximability, for any $\varepsilon > 0$, unless $\mathrm{P} = \mathrm{NP}$. Indyk et al. [30] gave a distributed $1/3$-approximation algorithm via composable coresets for diversity and coverage maximization. Borassi et al. [12] designed a $1/5$-approximation algorithm in the sliding window model. For asymmetric distances, Kumpulainen et al. [38] recently designed a $1/(6k)$-approximation algorithm. There are also many related works on mixed-integer programming formulations, heuristics, and applications to drug discovery [20, 52, 55, 60, 37, 40, 61, 42].

Bhaskara et al. [11] studied the *sum-min diversity* problem $\max_{S \subseteq V : |S| \leq k} \sum_{u \in S} \mathrm{dist}(u, S \setminus \{u\})$, providing the first constant-factor approximation algorithm with a $1/8$ guarantee. Their algorithm is based on a novel linear-programming relaxation and generalizes to matroid constraints. They also proved the first inapproximability result of $1/2$ under the planted clique assumption.

Diversity maximization has also recently been studied with *fairness constraints*. Given a partition of the points into $m$ groups, they ensure $k_i \in [\ell_i, u_i]$ points are selected from each group $i \in [m]$, subject to $|S| = k = \sum_{i=1}^{m} k_i$. This line of work can be categorized as studying fair max-min [46, 2, 61], sum-min [11, 41], and max-sum [1, 41] diversity objectives.

**Data sampling.** In the realm of dataset curation and active learning, there have been several lines of work that study the combination of utility and diversity terms. Greedy coreset methods based on $k$-center have been successfully used for data selection with and without utility terms [50, 57]. Ash et al. [6] use $k$-means++ seeding over the gradient space of a model to balance uncertainty and diversity. Wei et al. [62] introduce several submodular objectives, e.g., facility location, and use them to diversify a set of uncertain examples in each active learning iteration. Citovsky et al. [18] cluster examples represented by embeddings extracted from the penultimate layer of a partially trained DNN, and use these clusters to diversify uncertain examples in each iteration. Our work differs from these and several others (e.g., Kirsch et al. [35], Zhdanov [63]) in that we directly incorporate the utility and diversity terms into the objective function.

## 2 Preliminaries

**Submodular function.** For any $g : 2^V \to \mathbb{R}_{\geq 0}$ and $S, T \subseteq V$, let $g(S \mid T) = g(S \cup T) - g(T)$ be the *marginal gain* of $g$ at $S$ with respect to $T$. A function $g$ is submodular if for every $S \subseteq T \subseteq V$ and $v \in V \setminus T$, we have $g(v \mid S) \geq g(v \mid T)$, where we overload the marginal gain notation for singletons. A submodular function $g$ is *monotone* if for every $S \subseteq T \subseteq V$, $g(S) \leq g(T)$.

**Max-min diversity.** For a set of $n$ points $V$ in a metric space, define the *max-min diversity* function as

$$\mathtt{div}(S) = \begin{cases} \min_{u,v \in S : u \neq v} \mathrm{dist}(u,v) & \text{if } |S| \geq 2, \\ \max_{u,v \in V} \mathrm{dist}(u,v) & \text{if } |S| \leq 1. \end{cases}$$

We take $\mathtt{div}(S)$ to be the diameter of $V$ if $|S| \leq 1$ so that it is monotone decreasing. We extend the distance function to take subsets $S \subseteq V$ as input in the standard way, i.e., $\mathrm{dist}(u, S) = \min_{v \in S} \mathrm{dist}(u, v)$, and define $\mathrm{dist}(u, \varnothing) = \infty$.

**MDMS problem statement.** For any nonnegative monotone submodular function $g : 2^V \to \mathbb{R}_{\geq 0}$ and $\lambda \geq 0$, let

$$f(S) = g(S) + \lambda \cdot \mathtt{div}(S). \tag{2}$$

The *max-min diversification with monotone submodular utility* (MDMS) problem is to maximize $f(S)$ subject to a cardinality constraint $k$:

$$S^* = \underset{S \subseteq V : |S| \leq k}{\arg\max} f(S).$$

Let $\mathrm{OPT} = f(S^*)$ denote the optimal objective value.

*Remark* 2.1. The objective function $f(S)$ is *not submodular* (see Appendix A for a counterexample).

**Intersection graph.** Let $G_d(V)$ be the *intersection graph* of $V$ for distance threshold $d$, i.e., with nodes $V$ and edges $E = \{(u,v) \in V^2 : u \neq v, \mathrm{dist}(u,v) < d\}$. For any independent set $S$ of $G_d(V)$ and pair of distinct nodes $u, v \in S$, we have $\mathrm{dist}(u,v) \geq d$.

## 2.1 Warm-up: Simple $0.387$-approximation algorithm

The objective function $f(S)$ is composed of two terms, each of which is easy to (approximately) optimize in isolation. For the submodular term $g(S)$, run the greedy algorithm to get $S_1$ satisfying

$$g(S_1) \geq (1 - 1/e) \cdot \max_{|S| \leq k} g(S)$$
$$\geq (1 - 1/e) \cdot g(S^*).$$

For the $\mathtt{div}(S)$ term, let $S_2$ be a pair of diametrical points in $V$. Then, return the better of the two solutions. This gives a $0.387$-approximation guarantee because, for any $0 \leq p \leq 1$, we have:

$$\begin{aligned}
\mathrm{ALG}_{\mathrm{simple}} &= \max\{f(S_1), f(S_2)\} \\
&\geq p \cdot g(S_1) + (1-p) \cdot \lambda \cdot \mathtt{div}(S_2) \\
&\geq p \cdot (1 - 1/e) \cdot g(S^*) + (1-p) \cdot \lambda \cdot \mathtt{div}(S^*) \\
&= \frac{e-1}{2e-1} \cdot \mathrm{OPT},
\end{aligned}$$

where we set $p = e/(2e-1)$ at the end by solving $p \cdot (1 - 1/e) = 1 - p$. One of our main goals is to improve over this baseline and prove complementary hardness of approximation results.

One of the most common algorithms for subset selection problems is the greedy algorithm that iteratively adds the item with maximum marginal value with respect to the objective function $f$. We show in Appendix B that the greedy algorithm does not provide a constant-factor approximation guarantee.

## 3 Algorithm

In this section, we present the GIST algorithm for the MDMS problem and prove that it achieves an approximation ratio of $1/2 - \varepsilon$. At a high level, GIST works by sweeping over multiple distance thresholds $d$ and calling a GreedyIndependentSet subroutine on the intersection graph $G_d(V)$ to find a high-valued maximal independent set. This is in contrast to the warm-up $0.387$-approximation algorithm, which only considers the two extreme thresholds $d \in \{0, d_{\max}\}$.

GreedyIndependentSet builds the set $S$ (starting from the empty set) by iteratively adding $v \in V \setminus S$ with the highest marginal gain with respect to the submodular utility $g$ while satisfying $\mathrm{dist}(v, S) \geq d$. This subroutine runs until either $|S| = k$ or $S$ is a maximal independent set of $G_d(V)$.

GIST computes multiple solutions and returns the one with maximum value $f(S)$. It first runs the classic greedy algorithm for monotone submodular functions to get an initial solution, which is equivalent to calling GreedyIndependentSet with distance threshold $d = 0$. Then, it considers the set of distance thresholds $D \leftarrow \{(1 + \varepsilon)^i \cdot \varepsilon d_{\max}/2 : (1 + \varepsilon)^i \leq 2/\varepsilon \text{ and } i \in \mathbb{Z}_{\geq 0}\}$. The set $D$ contains a threshold close to the target $d^*/2$ where $d^* = \mathtt{div}(S^*)$.[2] For each $d \in D$, GIST calls GreedyIndependentSet$(V, g, d, k)$ to find a set $T$ of size at most $k$. If $T$ has a larger objective value than the best solution so far, it updates $S \leftarrow T$. After iterating over all thresholds in $D$, it returns the highest-value solution among all candidates.

**Theorem 3.1.** *For any $\varepsilon > 0$, GIST outputs a set $S \subseteq V$ with $|S| \leq k$ and $f(S) \geq (1/2 - \varepsilon) \cdot \mathrm{OPT}$ using $O(nk \log_{1+\varepsilon}(1/\varepsilon))$ submodular value oracle queries.*

The main building block in our design and analysis of GIST is the following lemma, which is inspired by the greedy $2$-approximation algorithm for metric $k$-center and adapted for monotone submodular utility functions.

**Lemma 3.2.** *Let $S_d^*$ be a maximum-value set of size at most $k$ with diversity at least $d$. In other words,*

$$S_d^* = \underset{S:|S| \leq k, \mathtt{div}(S) \geq d}{\arg\max} g(S).$$

*Let $T$ be the output of GreedyIndependentSet$(V, g, d', k)$. If $d' < d/2$, then $g(T) \geq g(S_d^*)/2$.*

---

[2]If this is not true, we are in the special case where $d^* \leq \varepsilon d_{\max}$, which we analyze separately.

**Algorithm 1** Max-min diversification with submodular utility via greedy weighted independent sets.

1: **function** GIST(points $V$, monotone submodular function $g : 2^V \to \mathbb{R}_{\geq 0}$, budget $k$, error $\varepsilon$)
2:      Initialize $S \leftarrow$ GreedyIndependentSet$(V, g, 0, k)$          ▷ classic greedy algorithm
3:      Let $d_{\max} = \max_{u,v \in V} \text{dist}(u, v)$ be the diameter of $V$
4:      Let $T \leftarrow \{u, v\}$ be two points such that $\text{dist}(u, v) = d_{\max}$
5:      **if** $f(T) > f(S)$ and $k \geq 2$ **then**
6:          Update $S \leftarrow T$
7:      Let $D \leftarrow \{(1+\varepsilon)^i \cdot \varepsilon d_{\max}/2 : (1+\varepsilon)^i \leq 2/\varepsilon \text{ and } i \in \mathbb{Z}_{\geq 0}\}$     ▷ distance thresholds
8:      **for** threshold $d \in D$ **do**
9:          Set $T \leftarrow$ GreedyIndependentSet$(V, g, d, k)$
10:         **if** $f(T) \geq f(S)$ **then**
11:             Update $S \leftarrow T$
         **return** $S$

1: **function** GreedyIndependentSet(points $V$, monotone submodular function $g : 2^V \to \mathbb{R}_{\geq 0}$, distance $d$, budget $k$)
2:      Initialize $S \leftarrow \varnothing$
3:      **for** $i = 1$ to $k$ **do**
4:          Let $C \leftarrow \{v \in V \setminus S : \text{dist}(v, S) \geq d\}$
5:          **if** $C = \varnothing$ **then**
6:             **return** $S$                    ▷ $S$ is a maximal independent set of $G_d(V)$
7:          Find $t \leftarrow \arg\max_{v \in C} g(v \mid S)$
8:          Update $S \leftarrow S \cup \{t\}$
         **return** $S$

*Proof.* Let $k' = |T|$ and $t_1, t_2, \ldots, t_{k'}$ be the points in $T$ in selection order. Let $B_i = \{v \in V : \text{dist}(t_i, v) < d'\}$ be the points in $V$ in the radius-$d'$ open ball around $t_i$. Since the distance between any pair of points in $B_i$ is at most $2d' < d$, each set $B_i$ contains at most one point in $S_d^*$.

First we construct an injective map $h : S_d^* \to T$. We say that $B_i$ *covers* a point $v \in V$ if $v \in B_i$. For each covered $s \in S_d^*$, we map it to $t_i$, where $i$ is the minimum index for which $B_i$ covers $s$. If there is an $s \in S_d^*$ not covered by any set $B_i$, then GreedyIndependentSet must have selected $k$ points, i.e., $|T| = k$. We map the uncovered points in $S_d^*$ to arbitrary points in $T$ while preserving the injective property.

Since $g$ is a monotone submodular function, we have

$$g(S_d^*) - g(T) \leq \sum_{s \in S_d^*} g(s \mid T).$$

We use $h$ to account for $g(s \mid T)$ in terms of the values we gained in set $T$. For any $s \in S_d^*$, let $T_s$ be the set of points added to set $T$ in algorithm GreedyIndependentSet right before the addition of $h(s)$. Since GreedyIndependentSet iteratively adds points and never deletes points from $T$, we know $T_s \subseteq T$. By submodularity, $g(s \mid T) \leq g(s \mid T_s)$. By the greedy nature of the algorithm, we also know that

$$g(h(s) \mid T_s) \geq g(s \mid T_s).$$

We note that the sum of the former terms, $g(h(s) \mid T_s)$, is at most $g(T)$ since $h$ is injective and $g$ is nondecreasing. Thus, we conclude that

$$g(S_d^*) - g(T) \leq \sum_{s \in S_d^*} g(s \mid T) \leq \sum_{s \in S_d^*} g(h(s) \mid T_s) \leq g(T),$$

which completes the proof. $\qquad\square$

Now that we have with this bicriteria approximation, we can analyze the approximation ratio of GIST.

*Proof of Theorem 3.1.* We first prove the oracle complexity of GIST. There are $1 + \log_{1+\varepsilon}(2/\varepsilon) = O(\log_{1+\varepsilon}(1/\varepsilon))$ thresholds in $D$. For each threshold, we call GreedyIndependentSet once. In each call for $k$ iterations, we find the maximum marginal-value point by scanning (in the worst case) all points. This requires at most $nk$ oracle calls yielding the overall oracle complexity of the algorithm.

Let $d^*$ be the minimum pairwise distance between points in $S^*$. The GIST algorithm iterates over a set of thresholds $D$. The definition of $D$ implies that at least one threshold $d$ is in the interval $[d^*/(2(1+\varepsilon)), d^*/2)$, unless $d^* \leq \varepsilon d_{\max}$. We deal with this special case later and focus on the case when such a $d$ exists.

Let $T$ be the output of GreedyIndependentSet for threshold value $d$. Since $d < d^*/2$, Lemma 3.2 implies that

$$g(T) \geq \frac{1}{2} \cdot g(S_{d^*}^*) \geq \frac{1}{2} \cdot g(S^*).$$

We also know from our choice of $d$ that

$$\mathtt{div}(T) \geq \frac{1}{2(1+\varepsilon)} \cdot \mathtt{div}(S^*).$$

Combining these inequalities gives us

$$f(T) \geq \frac{1}{2(1+\varepsilon)} \cdot \mathrm{OPT} > \left(\frac{1}{2} - \varepsilon\right) \cdot \mathrm{OPT}.$$

It remains to prove the claim for the case when $d^* \leq \varepsilon d_{\max}$. GIST considers two initial feasible sets and picks the better of the two as the initial value for $T$. The first set is the classic greedy solution [47] for the monotone submodular function $g(S)$, and ignores the diversity term. It follows that

$$f(T) \geq \left(1 - \frac{1}{e}\right) \cdot g(S^*) \geq \left(1 - \frac{1}{e}\right) \cdot (\mathrm{OPT} - \lambda \cdot \varepsilon d_{\max}). \tag{3}$$

The second set contains two points with pairwise distance $d_{\max}$, and ignores the submodular term. This yields the lower bound

$$f(T) \geq \lambda \cdot d_{\max} \implies -\lambda \cdot d_{\max} \geq -f(T). \tag{4}$$

Combining (3) and (4), we get a final bound of $f(T) \geq (1 - \frac{1}{e}) \cdot \frac{\mathrm{OPT}}{1+\varepsilon}$, which completes the proof. $\square$

### 3.1 Linear utility

If $g(S) = \sum_{v \in S} w(v)$ is a linear utility with nonnegative weights $w : V \to \mathbb{R}_{\geq 0}$, then Theorem 3.1 gives us a $(1/2 - \varepsilon)$-approximation since this is a special case of submodularity. However, GIST offers a *stronger approximation ratio* under this assumption.

**Theorem 3.3.** *Let $g(S) = \sum_{v \in S} w(v)$ be a linear function. For any $\varepsilon > 0$, GIST returns $S \subseteq V$ with $|S| \leq k$ and $f(S) \geq (2/3 - \varepsilon) \cdot \mathrm{OPT}$.*

We defer the proof to Appendix C.1, but explain the main differences. For linear functions, Lemma 3.2 can be strengthened to show that GreedyIndependentSet outputs a set $T$ such that $g(T) \geq g(S_d^*)$, for any $d' < d/2$. Then, for some $d \in D$, we get $\mathrm{ALG} \geq \max\{g(S^*) + \lambda \cdot d^*/(2(1+\varepsilon)), \lambda \cdot d^*\}$, and the optimal convex combination of these two lower bounds gives the approximation ratio.

## 4 Hardness of approximation

We begin by summarizing our hardness results for MDMS. Assuming $P \neq NP$, we prove that:

**Submodular utility.** There is no polynomial-time $0.5584$-approximation algorithm if $g$ is a nonnegative, monotone submodular function.

**Linear utility.**

- There is no polynomial-time $(2/3 + \varepsilon)$-approximation algorithm if $g$ is linear, for any $\varepsilon > 0$, for general distance metrics.
- APX-completeness for linear utility functions even in the Euclidean metric, i.e., there is no *polynomial-time approximation scheme* (PTAS) for this problem.

### 4.1 General metric spaces

We build on the set cover hardness instances originally designed by Feige et al. [23, 25, 24], and then further engineered by Kapralov et al. [32]. We present the instances in [32, Section 2] as follows:

**Hardness of max $k$-cover.** For any constants $c, \varepsilon > 0$ and a given collection/family of sets $\mathcal{F}$ partitioned into groups $\mathcal{F}_1, \mathcal{F}_2, \cdots, \mathcal{F}_k$, it is NP-hard to distinguish between the following two cases:

- YES case: there exists $k$ disjoint sets $S_i$, one from each $\mathcal{F}_i$, whose union covers the entire universe.
- NO case: for any $\ell \leq c \cdot k$ sets, their union covers at most a $(1 - (1 - 1/k)^\ell + \varepsilon)$-fraction of the universe.

The construction above can be done such that each set $S_i \in \mathcal{F} = \bigcup_{r=1}^{k} \mathcal{F}_r$ has the same size. We note that the parameter $k$ should be larger than any constant we set since there is always an exhaustive search algorithm with running time $|\mathcal{F}|^k$ to distinguish between the YES and NO cases. Therefore, we can assume, e.g., that $k \geq 1/\varepsilon$.

We use this max $k$-cover instance to prove hardness of approximation for the MDMS problem.

**Theorem 4.1.** *It is* NP*-hard to approximate* MDMS *within a factor of* $\frac{2(1 - 1/e)}{2(1 - 1/e) + 1} + \delta < 0.55836 + \delta$, *for any $\delta > 0$.*

*Proof sketch.* We reduce an instance of the max $k$-cover problem to MDMS to demonstrate how the inapproximability result translates. For any pair of sets in the collection $\mathcal{F}$, their distance is defined to be 2 if and only if they are disjoint; otherwise, it is 1. Consequently, a selected sub-collection of $\mathcal{F}$ achieves the higher diversity score of 2 if and only if all chosen sets are mutually disjoint.

Given that all sets in $\mathcal{F}$ are of uniform size and considering the upper bound in the NO case on the union coverage for any collection of sets, a polynomial-time algorithm can only find $O(\varepsilon k)$ pairwise disjoint sets. Thus, if an algorithm aims for a diversity score of 2 (by selecting only disjoint sets), it forgoes most of the potential value from the submodular coverage function in the MDMS formulation. On the other hand, if the algorithm settles for the lower diversity score of 1, it still cannot cover more than an approximate fraction $1 - (1 - 1/k)^k + \varepsilon \approx 1 - 1/e + \varepsilon$ of the universe.

Combining these two upper bounds on any polynomial-time algorithm's performance with the fact that, in the YES case, the optimal solution covers the entire universe and achieves diversity score 2, establishes the claimed hardness of approximation. $\square$

## 4.2 Linear utility

For the special case of linear utility functions, we prove a tight hardness result to complement our $2/3$-approximation guarantee in Theorem 3.3.

**Theorem 4.2.** *For any $\varepsilon > 0$, there is no polynomial-time $(2/3 + \varepsilon)$-approximation algorithm for the* MDMS *problem if $g$ is a linear function, unless* P = NP.

We defer the proof to Appendix D.2. Our construction builds on the work of Håstad [28] and Zuckerman [64], which shows that the max clique problem does not admit an efficient $n^{1 - \theta}$-approximation algorithm, for any constant $\theta > 0$.

## 4.3 Euclidean metric

Our final result is for the Euclidean metric, i.e., $S \subseteq \mathbb{R}^d$ and $\text{dist}(\mathbf{x}, \mathbf{y}) = \|\mathbf{x} - \mathbf{y}\|_2$. We build on a result of Alimonti and Kann [4] showing that the size of a max independent set in a bounded-degree graph cannot be approximated to within a constant factor $1 - \varepsilon$, for any $\varepsilon > 0$, unless P = NP.

**Lemma 4.3** (Alimonti and Kann [4, Theorem 3.2]). *The maximum independent set problem for graphs with degree at most 3 is* APX*-complete.*

Our second ingredient is an embedding function $h_G(v)$ that encodes graph adjacency in Euclidean space.

**Lemma 4.4.** *Let $G = (V, E)$ be a simple undirected graph with $n = |V|$, $m = |E|$, and max degree $\Delta$. There exists an embedding $h_G : V \to \mathbb{R}^{n+m}$ such that if $\{u, v\} \in E$ then*

$$\|h_G(u) - h_G(v)\|_2 \leq 1 - \frac{1}{2(\Delta + 1)},$$

*and if $\{u, v\} \notin E$ then $\|h_G(u) - h_G(v)\|_2 = 1$.*

**Theorem 4.5.** MDMS *is* APX-*complete for the Euclidean metric if g is a linear function.*

We sketch the main idea below and defer proofs to Appendix D. Let $\mathcal{I}(G)$ be the set of independent sets of $G$. Using Lemma 4.4, for any set of nodes $S \subseteq V$ with $|S| \geq 2$ in a graph $G$ with max degree $\Delta \leq 3$, we have the property:

- $S \in \mathcal{I}(G) \implies \text{div}(S) = 1$;
- $S \notin \mathcal{I}(G) \implies \text{div}(S) \leq 1 - \frac{1}{2(\Delta+1)} \leq 1 - \frac{1}{8}$.

The gap is at least $\frac{1}{8}$ for all such graphs (i.e., a universal constant), so setting $\lambda = 1$ allows us to prove that MDMS inherits the APX-completeness of bounded-degree max independent set.

## 5 Experiments

### 5.1 Warm-up: Synthetic dataset

We first compare GIST against baseline methods on a simple MDMS task with a normalized budget-additive utility function (i.e., monotone submodular) and a set of weighted Guassian points.

**Setup.** Generate $n = 1000$ points $\mathbf{x}_i \in \mathbb{R}^d$, for $d = 64$, where each $\mathbf{x}_i \sim \mathcal{N}(\mathbf{0}, \mathbf{I}_d)$ is i.i.d. and assigned a uniform continuous weight $w_i \sim \mathcal{U}_{[0,1]}$. For $\alpha, \beta \in [0, 1]$, the objective function trades off between the average (capped) utility of the points and their min-distance diversity reward:

$$f(S) = \alpha \cdot \min\left\{\frac{1}{k}\sum_{i \in S} w_i, \, \beta\right\} + (1 - \alpha) \cdot \text{div}(S).$$

We consider three baseline methods: random, simple, and greedy. random selects $k$ random points, permutes them, and returns the best prefix since the objective function $f(S)$ is non-monotone. simple is the 0.387-approximation algorithm in Section 2.1. greedy builds $S$ one point at a time by selecting the point with index $i_t^* = \arg\max_{i \in V \setminus S_{t-1}} f(S_{t-1} \cup \{i\}) - f(S_{t-1})$ at each step $t \in [k]$, and returns the best prefix $S = \arg\max_{t \in [k]} f(S_t)$. GIST considers all possible $D \leftarrow \{\text{dist}(u, v)/2 : u, v \in V\}$ since $n = 1000$, which yields an exact $\frac{2}{3}$-approximation ratio.

**Results.** We plot the values of $f(S_{\text{ALG}})$ for the baseline methods and GIST, for each $k \in [n]$, in Figure 1. GIST dominates simple, greedy, and random in all cases. greedy performs poorly for $k \geq 100$, which is surprising since it is normally a competitive method for subset selection tasks. simple beats greedy for mid-range values of $k$, i.e., $k \in [250, 900]$, but it is always worse than GIST. In Appendix E.1, we sweep over $\alpha$ and $\beta$, and show how these hyperparameters affect the $f(S_{\text{ALG}})$ plots. Finally, we remark that the plots in Figure 1 are decreasing in $k$ because (i) we use a *normalized* budget-additive utility function, and (ii) $\text{div}(S)$ is a monotone-decreasing set function.

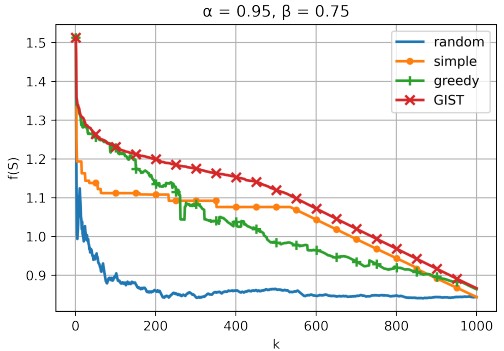

*Figure 1:* $f(S_{\text{ALG}})$ for baseline methods and GIST, for each cardinality constraint $k \in [n]$, on synthetic data with $n = 1000$, $\alpha = 0.95$, and $\beta = 0.75$.

### 5.2 Image classification

Our ImageNet data sampling experiment compares the top-1 image classification accuracy achieved by different single-shot subset selection algorithms.

**Setup.** We use the standard vision dataset ImageNet [54] containing ~1.3 million images and 1000 classes. We select 10% of the images uniformly at random and use them to train an initial ResNet-56 model $\boldsymbol{\theta}_0$ [29]. Then we use the model $\boldsymbol{\theta}_0$ to compute a 2048-dimensional unit-length embedding $\boldsymbol{e}_i$

*Table 1:* Top-1 classification accuracy (%) on ImageNet for different single-shot data downsampling algorithms. The cardinality constraint $k$ is expressed as a percent of the ~1.3 million examples. The results are the average of three trials and the top performance for each $k$ is shown in bold.

| $k$ (%) | random | margin | $k$-center | submod | GIST-margin | GIST-submod |
|---|---|---|---|---|---|---|
| 30 | $66.23 \pm 0.15$ | $65.97 \pm 0.33$ | $66.71 \pm 0.57$ | $66.52 \pm 0.21$ | $66.90 \pm 0.19$ | $\mathbf{67.24} \pm 0.06$ |
| 40 | $69.17 \pm 0.12$ | $69.73 \pm 0.38$ | $70.06 \pm 0.06$ | $70.71 \pm 0.24$ | $70.51 \pm 0.12$ | $\mathbf{70.76} \pm 0.35$ |
| 50 | $71.05 \pm 0.16$ | $72.33 \pm 0.09$ | $73.01 \pm 0.10$ | $72.69 \pm 0.15$ | $72.69 \pm 0.34$ | $\mathbf{73.15} \pm 0.21$ |
| 60 | $72.49 \pm 0.28$ | $73.43 \pm 0.06$ | $73.60 \pm 0.31$ | $74.20 \pm 0.08$ | $\mathbf{74.34} \pm 0.01$ | $74.30 \pm 0.27$ |
| 70 | $73.70 \pm 0.22$ | $74.49 \pm 0.22$ | $74.24 \pm 0.23$ | $75.24 \pm 0.30$ | $\mathbf{75.41} \pm 0.23$ | $75.32 \pm 0.19$ |
| 80 | $74.42 \pm 0.39$ | $75.01 \pm 0.05$ | $75.17 \pm 0.11$ | $75.91 \pm 0.07$ | $\mathbf{75.96} \pm 0.28$ | $75.45 \pm 0.16$ |
| 90 | $75.16 \pm 0.13$ | $75.11 \pm 0.13$ | $75.00 \pm 0.38$ | $75.84 \pm 0.10$ | $\mathbf{76.12} \pm 0.33$ | $76.03 \pm 0.02$ |

and uncertainty score for each example $\boldsymbol{x}_i$. The *margin-based* uncertainty score of $\mathbf{x}_i$ is given by $u_i = 1 - (\Pr(y = b \mid \mathbf{x}_i; \boldsymbol{\theta}_0) - \Pr(y = b' \mid \mathbf{x}_i; \boldsymbol{\theta}_0))$, which measures the difference between the probability of the best predicted class label $b$ and second-best label $b'$ for an example. Finally, we use the fast maximum inner product search of Guo et al. [27] to build a $\Delta$-nearest neighbor graph $G$ in the embedding space using $\Delta = 100$ and cosine distance, i.e., $\text{dist}(\boldsymbol{x}_i, \boldsymbol{x}_j) = 1 - \boldsymbol{e}_i \cdot \boldsymbol{e}_j$. We present all model training hyperparameters in Appendix E.

We compare GIST with several state-of-the-art benchmarks:

- random: We draw samples from the dataset uniformly at random without replacement. This is a simple and lightweight approach that promotes diversity in many settings and provides good solutions.

- margin [53]: Margin sampling selects the top-$k$ points using the uncertainty scores $u_i$, i.e., based on how hard they are to classify. It is not incentivized to output a diverse set of training examples.

- $k$-center [57]: We run the classic greedy algorithm for $k$-center on $G$. We take the distance between non-adjacent nodes to be the max distance among all pairs of adjacent nodes in $G$.

- submod: We select a subset by greedily maximizing the submodular objective function

$$g(S) = \alpha_s \sum_{i \in S} u_i - \beta_s \sum_{i,j \in S} s(i,j), \tag{5}$$

subject to the constraint $|S| \leq k$, where $s(i,j) = 1 - \text{dist}(\boldsymbol{x}_i, \boldsymbol{x}_j)$ is the cosine similarity between adjacent nodes in $G$. Similar pairwise-diversity submodular objective functions have also been used in [45, 21, 31, 36, 49]. This is a *different diversity objective* than $\text{div}(S)$ that keeps $g(S)$ submodular but allows it to be non-monotone. We tuned for best performance by selecting $\alpha_s = 0.9$ and $\beta_s = 0.1$.

Finally, we bootstrap the margin and submod objectives with MDMS and run GIST with $\varepsilon = 0.05$:

- GIST-margin: Let $f(S) = \alpha \cdot \sum_{i \in S} u_i + (1-\alpha) \cdot \text{div}(S)$ for $\alpha \in [0, 1]$. This uses the same *linear utility function* as margin sampling. We optimize for performance and set $\alpha = 0.9$.

- GIST-submod: Let $f(S) = \alpha \cdot g(S) + (1-\alpha) \cdot \text{div}(S)$, where $g(S)$ is the same submodular function in (5) with $\alpha_s = 0.9$ and $\beta_s = 0.1$. We optimize for performance and set $\alpha = 0.95$.

**Results.** We run each sampling algorithm with cardinality constraint $k$ on the full dataset to get a subset of examples that we then use to train a new ResNet-56 model. We report the average top-1 classification accuracy of these models in Table 1. GIST with margin or submodular utility is superior to all baselines. Interestingly, there is a cut-over value of $k$ where the best algorithm switches from GIST-submod to GIST-margin. We also observe that GIST-submod and GIST-margin outperform submod and margin, respectively. This demonstrates how $\text{div}(S)$ encourages diversity in the set of sampled points and improves downstream model quality. The running time of margin and submod is 3–4 minutes per run on average. GIST is similar to the margin or submodular algorithms for a given distance threshold $d$. The end-to-end running time is dominated by training ImageNet models, which takes more than a few hours even with several accelerators (e.g., GPU/TPU chips).

## Conclusion

We introduce a novel subset selection problem called MDMS that combines the utility of the selected points (modeled as a monotone submodular function $g$) with the $\text{div}(S) = \min_{u,v \in S:u \neq v} \text{dist}(u,v)$ diversity objective. We design and analyze the GIST algorithm, which achieves a $1/2$-approximation guarantee by solving a series of maximal-weight independent set instances on intersection graphs with the GreedyIndependentSet bicriteria-approximation algorithm. We complement GIST with a $0.5584$ hardness of approximation. It is an interesting open theory problem to close the gap between the $0.5$ approximation ratio and $0.5584$ inapproximability. For linear utilities, we show that GIST achieves a $2/3$-approximation and that it is NP-hard to find a $(2/3 + \varepsilon)$-approximation, for any $\varepsilon > 0$.

Our empirical study starts by comparing GIST to existing methods for MDMS on a simple synthetic task to show the shortcomings of baseline methods (in particular the greedy algorithm). Then we compare the top-1 image classification accuracy of GIST and state-of-the-art data sampling methods for ImageNet, demonstrating the benefit of optimizing the trade-off between a submodular utility and max-min diversity.

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

## A  $f(S)$ is not submodular

Consider the instance on four points $N = \{a, b, c, d\} \subseteq \mathbb{R}$ where $a = 0$, $b = 1$, and $c = d = 2$ with the (one-dimensional) Euclidean metric, i.e., all the points are collinear. Let $g(S) = 0$ for all $S \subseteq V$. For $f(S)$ to be submodular, it must hold that for every $S \subseteq T \subseteq N$ and $x \in N \setminus T$,

$$f(S \cup \{x\}) - f(S) \geq f(T \cup \{x\}) - f(T).$$

However, if $x = b$, $S = \{a, c\}$ and $T = \{a, c, d\}$, we have:

$$f(S \cup \{x\}) - f(S) = f(\{a, b, c\}) - f(\{a, c\}) = 1 - 2 = -1$$
$$f(T \cup \{x\}) - f(T) = f(\{a, b, c, d\}) - f(\{a, c, d\}) = 0 - 0 = 0,$$

so $f(S)$ is not submodular.

In this instance, $f(S)$ is not monotone. However, a similar monotone but still not submodular $f(S)$ can be defined by setting $g(S) = \sum_{v \in S} w(v)$ where $w(v) = 2$ for every $v \in N$.

## B  Greedy does not give a constant-factor approximation guarantee

The objective function $f$ is highly non-monotone since the diversity term can decrease as we add items. Consequently, the standard greedy algorithm, which rejects any item with a negative marginal gain, can have an arbitrarily poor performance.

We demonstrate this with a hard instance. Let the submodular part of the objective be $g(S) = |S|$. For the diversity component, define the distance between a specific pair $(u, v)$ to be $\text{dist}(u, v) = 2 + 2\varepsilon$, while for all other pairs $(x, y) \neq (u, v)$, we have $\text{dist}(x, y) = 1 + \varepsilon$. The greedy algorithm first selects the set $\{u, v\}$, achieving a value of $f(\{u, v\}) = g(\{u, v\}) + \text{dist}(u, v) = 4 + 2\varepsilon$. The algorithm then terminates because adding any subsequent item gives a submodular gain of 1 but causes a diversity loss of $1 + \varepsilon$, resulting in a negative marginal gain. An optimal solution of size $k$, however, can achieve a value of at least $k$ from the submodular term alone. The resulting approximation ratio for greedy is at most $(4 + 2\varepsilon)/k$, which approaches 0 as $k$ grows. Thus, greedy offers no constant-factor approximation guarantee. Furthermore, modifying greedy to accept items with negative marginal value does not solve this problem since one can construct instances where an optimal solution value is dominated by a diversity term such that selecting any set of size $k$ reduces the diversity term to zero.

## C  Missing analysis for Section 3

### C.1  Proof of Theorem 3.3

The following result is a tighter analysis of the bicriteria approximation of `GreedyIndependentSet` (Lemma 3.2) if $g(S)$ is a linear function. This is the key ingredient for improving the approximation ratio of `GIST` to $2/3 - \varepsilon$.

**Lemma C.1.** *Let $g(S) = \sum_{v \in S} w(v)$ be a linear function with nonnegative weights $w : V \to \mathbb{R}_{\geq 0}$. Let $S_d^*$ be a max-weight independent set of the intersection graph $G_d(V)$ of size at most $k$. If $T$ is the output of* `GreedyIndependentSet`$(V, g, d', k)$*, for $d' \leq d/2$, then $w(T) \geq w(S_d^*)$.*

*Proof.* Let $k' = |T|$ and $t_1, t_2, \ldots, t_{k'}$ be the points in $T$ in the order that `GreedyIndependentSet` selected them. Let $B_i = \{v \in V : \text{dist}(t_i, v) < d'\}$ be the points in $V$ contained in the radius-$d'$ open ball around $t_i$. First, we show that each $B_i$ contains at most one point in $S_d^*$. If this is not true, then some $B_i$ contains two different points $u, v \in S_d^*$. Since $\text{dist}(\cdot, \cdot)$ is a metric, this means

$$\text{dist}(u, v) \leq \text{dist}(u, t_i) + \text{dist}(t_i, v)$$
$$< d' + d'$$
$$\leq d/2 + d/2$$
$$= d,$$

which contradicts the assumption that $S_d^*$ is an independent set of $G_d(V)$. Note that it is possible to have $B_i \cap B_j \neq \varnothing$, for $i \neq j$, since these balls consider all points in $V$.

Now let $C_i \subseteq V$ be the set of uncovered points (by the open balls) that become covered when `GreedyIndependentSet` selects $t_i$. Concretely, $C_i = B_i \setminus (B_1 \cup \cdots \cup B_{i-1})$. Each $C_i$ contains at most one point in $S_d^*$ since $|B_i \cap S_d^*| \leq 1$. Moreover, if $s \in C_i \cap S_d^*$, then $w(t_i) \geq w(s)$ because the points are sorted in non-increasing order and selected if uncovered.

Let $A = C_1 \cup \cdots \cup C_{k'}$ be the set of points covered by the algorithm. For each point $s \in S_d^* \cap A$, there is exactly one covering set $C_i$ corresponding to $s$. It follows that

$$\sum_{s \in S_d^* \cap A} w(s) \leq \sum_{i \in [k']:S_d^* \cap C_i \neq \varnothing} w(t_i). \tag{6}$$

It remains to account for the points in $S_d^* \setminus A$. If we have any such points, then $|T| = k$ since the points in $S_d^* \setminus A$ are uncovered at the end of the algorithm. Further, for any $t_i \in T$ and $s \in S_d^* \setminus A$, we have $w(t_i) \geq w(s)$ since $t_i$ was selected and the points are sorted by non-increasing weight. Therefore, we can assign each $s \in S_d^* \setminus A$ to a unique $C_i$ such that $C_i \cap S_d^* = \varnothing$. It follows that

$$\sum_{s \in S_d^* \setminus A} w(s) \leq \sum_{i \in [k']:S_d^* \cap C_i = \varnothing} w(t_i). \tag{7}$$

Adding the two sums together in (6) and (7) completes the proof. $\qquad\square$

**Theorem 3.3.** *Let $g(S) = \sum_{v \in S} w(v)$ be a linear function. For any $\varepsilon > 0$, `GIST` returns $S \subseteq V$ with $|S| \leq k$ and $f(S) \geq (2/3 - \varepsilon) \cdot \mathrm{OPT}$.*

*Proof of Theorem 3.3.* Let $d^*$ be the minimum distance between two distinct points in $S^*$. There are two cases: $d^* \leq \varepsilon d_{\max}$ and $d^* > \varepsilon d_{\max}$. In the first case, outputting the $k$ heaviest points (Line 2 of `GIST`) yields a $(1 - \varepsilon)$-approximation. To see this, first observe that

$$\mathrm{OPT} \geq \lambda \cdot d_{\max} \geq \lambda \cdot \frac{d^*}{\varepsilon} \implies \varepsilon \cdot \mathrm{OPT} \geq \lambda \cdot d^*.$$

The sum of the $k$ heaviest points upper bounds $g(S^*)$, so we have

$$\mathrm{ALG} \geq g(S^*) = \mathrm{OPT} - \lambda \cdot d^* \geq (1 - \varepsilon) \cdot \mathrm{OPT}.$$

Now we consider the case where $d^* > \varepsilon d_{\max}$. `GIST` tries a threshold $d \in [d^*/(2(1 + \varepsilon)), d^*/2)$, so Lemma C.1 implies that `GreedyIndependentSet`$(V, g, d, k)$ outputs a set $T$ such that

$$f(T) \geq g(S^*) + \lambda \cdot d \geq g(S^*) + \lambda \cdot \frac{d^*}{2(1 + \varepsilon)}. \tag{8}$$

The max-diameter check on Lines 3–6 give us the lower bound

$$\mathrm{ALG} \geq \lambda \cdot d_{\max} \geq \lambda \cdot d^*. \tag{9}$$

Combining (8) and (9), the following inequality holds for any $0 \leq p \leq 1$:

$$\mathrm{ALG} \geq p \cdot \left[ g(S^*) + \lambda \cdot \frac{d^*}{2(1 + \varepsilon)} \right] + (1 - p) \cdot \lambda \cdot d^*$$

$$= p \cdot g(S^*) + \left( 1 - p + \frac{p}{2(1 + \varepsilon)} \right) \cdot \lambda \cdot d^*.$$

To maximize the approximation ratio as $\varepsilon \to 0$, we set $p = 2/3$ by solving $p = 1 - p/2$. Therefore,

$$\mathrm{ALG} \geq \frac{2}{3} \cdot g(S^*) + \left( 1 - \frac{2}{3} + \frac{1}{3(1 + \varepsilon)} \right) \cdot \lambda \cdot d^* \tag{10}$$

$$= \frac{2}{3} \cdot g(S^*) + \frac{1}{3} \left( 1 + \frac{1}{1 + \varepsilon} \right) \cdot \lambda \cdot d^*$$

$$\geq \frac{2}{3} \cdot g(S^*) + \frac{1}{3}(2 - \varepsilon) \cdot \lambda \cdot d^*$$

$$\geq \left( \frac{2}{3} - \varepsilon \right) \cdot \mathrm{OPT},$$

which completes the proof. $\qquad\square$

# D Missing analysis for Section 4

## D.1 Proof of Theorem 4.1

**Theorem 4.1.** *It is* NP*-hard to approximate* MDMS *within a factor of* $\frac{2(1-1/e)}{2(1-1/e)+1} + \delta < 0.55836 + \delta$, *for any* $\delta > 0$.

*Proof.* We construct our instance based on the collection of sets above as follows. We set $\varepsilon = \min\{\delta, \delta^2/6\}$ and $c = 1$. As mentioned above, we can assume that $k > 1/\varepsilon + 1$ since there is always a brute force polynomial time algorithm for constant $k$ to distinguish between YES and NO instances.

Suppose there are $n$ sets $S_1, S_2, \cdots, S_n$ in the collection $\mathcal{F}$. We represent these $n$ sets with $n$ points in the MDMS instance. We overload the notation to show the corresponding point with $S_i$ too. For any subset of sets/points $T$, the submodular value $g(T)$ is defined as the cardinality of the union of corresponding sets, i.e., the *coverage submodular function*. In other words, $g(T) = |\bigcup_{S_i \in T} S_i|$.

The distances between points/sets are either $d$ or $2d$. If two sets are disjoint and belong to two separate partitions $\mathcal{F}_r$ and $\mathcal{F}_{r'}$, their distance is $2d$. Otherwise, we set their distance to $d$. So for any two sets $S, S'$ belonging to the same group $\mathcal{F}_r$, their distance is set to $d$. Also, for any two sets with nonempty intersection, we set their distance to $d$ too. Any other pair of points will have distance $2d$.

We set $d$ to be $(1 - 1/e)U$ where $U$ is the cardinality of the union of all sets, i.e., $U = |\bigcup_{S \in \mathcal{F}} S|$. Finally, we set $\lambda = 1$ to complete the construction of the MDMS instance.

In the YES case, the optimum solution is the family of $k$ disjoint sets from different groups that cover the entire universe. The value of optimum in this case is $2d + U = (2(1 - 1/e) + 1)U$.

The algorithm has two possibilities: (case a) the algorithm gives up on the diversity objective and selects $k$ sets with minimum distance $d$, or (case b) it aims for a diversity term of $2d$. In case (a), the submodular value is at most $(1 - (1 - 1/k)^k + \varepsilon)U$ because of the property of the NO case. We note that if the algorithm finds a set of $k$ sets with union size above this threshold, we can conclude that we have a YES instance, which contradicts the hardness result.

The limit of the upper bound for the submodular value as $k$ goes to infinity is $1 - 1/e + \varepsilon$. We use its expansion series to derive:

$$\left(1 - \frac{1}{k}\right)^k \geq \frac{1}{e} - \frac{1}{2ek} - \frac{5}{24ek^2} - \cdots$$

The sequence of negative terms declines in absolute value with a rate of at least $1/k$. Thus, the absolute value of their total sum is at most the first deductive term $1/2ek$ times $1/(1 - 1/k) = k/(k - 1)$, which gives the simpler bound:

$$\left(1 - \frac{1}{k}\right)^k \geq \frac{1}{e} - \frac{1}{2e(k - 1)}.$$

Since $k - 1$ is at least $1/\varepsilon$, the submodular value in case (a) does not exceed:

$$\left(1 - \frac{1}{e} + \frac{1}{2e(k - 1)} + \varepsilon\right)U \leq \left(1 - \frac{1}{e} + 2\varepsilon\right)U.$$

Recall that $d = (1 - 1/e)U$, so the ratio of what the algorithm achieves in case (a) and the optimum solution of YES case is at most:

$$\frac{(1 - 1/e + 2\varepsilon)U + d}{U + 2d} = \frac{2(1 - 1/e) + 2\varepsilon}{2(1 - 1/e) + 1}$$
$$\leq \frac{2(1 - 1/e)}{2(1 - 1/e) + 1} + \delta.$$

In case (b), the algorithm is forced to pick only disjoint sets to maintain a minimum distance of $2d$. This means if the algorithm picks $\ell$ sets, their union has size $\ell \cdot U/k$. This is true because all sets in $\mathcal{F}$ have the same size and we know in the YES case, the union of $k$ disjoint sets covers the entire

universe hence each set has size $U/k$. For the special case of $\ell = 1$, we know that only a $1/k < \varepsilon$ fraction of universe is covered. We upper bound the covered fraction in terms of $\varepsilon$ for the other cases. The property of the NO case implies the following upper bound on $\ell$:

$$\frac{\ell}{k} \leq 1 - \left(1 - \frac{1}{k}\right)^{\ell} + \varepsilon.$$

We use the binomial expansion of $(1 - 1/k)^{\ell}$ and note that each negative term exceeds its following positive term in absolute value. This is true because of the cardinality constraint $\ell \leq k$. Therefore,

$$\left(1 - \frac{1}{k}\right)^{\ell} \geq 1 - \frac{\ell}{k} + \frac{\ell(\ell-1)}{2k^2} - \frac{\ell(\ell-1)(\ell-2)}{6k^3}$$

$$\geq 1 - \frac{\ell}{k} + \frac{\ell(\ell-1)}{3k^2}$$

$$\geq 1 - \frac{\ell}{k} + \frac{\ell^2}{6k^2},$$

where the second to last inequality holds since $\ell - 2 < k$ and the last inequality holds because $\ell \geq 2$ and consequently $\ell - 1 \geq \ell/2$. We can now revise the initial inequality:

$$\frac{\ell}{k} \leq (1 - (1 - 1/k)^{\ell} + \varepsilon)$$

$$\leq 1 - 1 + \frac{\ell}{k} - \frac{\ell^2}{6k^2} + \varepsilon \implies \frac{\ell^2}{6k^2} \leq \varepsilon.$$

Thus, the fraction of the universe that the algorithm covers, namely $\ell/k$, is at most $\sqrt{6\varepsilon} \leq \delta$.

In case (b), the ratio of what the algorithm achieves, and the optimum solution of YES case is at most:

$$\frac{\delta \cdot U + 2d}{U + 2d} = \frac{2(1 - 1/e) + \delta}{2(1 - 1/e) + 1} \leq \frac{2(1 - 1/e)}{2(1 - 1/e) + 1} + \delta.$$

This concludes the proof in both cases (a) and (b). □

### D.2 Proof of Theorem 4.2

**Theorem 4.2.** *For any $\varepsilon > 0$, there is no polynomial-time $(2/3 + \varepsilon)$-approximation algorithm for the* MDMS *problem if $g$ is a linear function, unless* $P = NP$.

*Proof.* First, recall that a clique is a subset of vertices in an undirected graph such that there is an edge between every pair of its vertices. Håstad [28] and Zuckerman [64] showed that the maximum clique problem does not admit an $n^{1-\theta}$-approximation for any constant $\theta > 0$, unless $NP = P$. This implies that there is no constant-factor approximation algorithm for maximum clique. In other words, for any constant $0 < \delta \leq 1$, there exists a graph $G$ and a threshold integer value $k$ such that it is NP-hard to distinguish between the following two cases:

- YES instance: graph $G$ has a clique of size $k$.

- NO instance: graph $G$ does not have a clique of size greater than $\delta k$.

We reduce this instance of the maximum clique decision problem to MDMS with objective function (2) as follows. Represent each vertex of graph $G$ with a point in our ground set. The distance between a pair of points is 2 if there is an edge between their corresponding vertices in $G$, and it is 1 otherwise.

Use the same threshold value of $k$ (in the YES and NO instance above) for the cardinality constraint on set $S$, and set each weight $w(v) = \alpha/k$ for some parameter $\alpha$ that we set later in the proof. We also set $\lambda = 1 - \alpha$. In a YES instance, selecting a clique of size $k$ as set $S$ results in the maximum possible value of the objective:

$$\text{OPT} = \alpha \cdot \frac{1}{k} \cdot k + (1 - \alpha) \cdot 2 = 2 - \alpha. \tag{11}$$

In a NO instance, the best objective value that can be achieved in polynomial-time is the maximum of the following two scenarios: (a) selecting $k$ points with minimum distance 1, or (b) selecting at most $\delta k$ vertices forming a clique with minimum distance 2. The maximum value obtained by any polynomial-time algorithm is then

$$\begin{aligned}
\text{ALG} &= \max\{\alpha + (1 - \alpha) \cdot 1, \alpha \cdot \delta + (1 - \alpha) \cdot 2\} \\
&= \max\{1, 2 - (2 - \delta)\alpha\}.
\end{aligned}$$

We make these two terms equal by setting $\alpha = 1/(2 - \delta)$. Thus, the gap between the maximum value any algorithm can achieve in the NO case and the optimum value in the YES case is

$$\frac{1}{2 - \alpha} = \frac{1}{2 - {}^1/(2 - \delta)} = \frac{2 - \delta}{3 - 2\delta}.$$

To complete the proof, it suffices to show that the ratio above is at most $2/3 + \varepsilon$. We separate the $2/3$ term as follows:

$$\frac{2 - \delta}{3 - 2\delta} = \frac{{}^2/3 \cdot (3 - 2\delta) + {}^\delta/3}{3 - 2\delta} = \frac{2}{3} + \frac{\delta}{9 - 6\delta}.$$

Therefore, we must choose a value of $\delta$ satisfying $\delta/(9 - 6\delta) \leq \varepsilon$. Since $\delta \leq 1$, the denominator $9 - 6\delta$ is positive. Equivalently, we want to satisfy:

$$\frac{9 - 6\delta}{\delta} = \frac{9}{\delta} - 6 \geq \frac{1}{\varepsilon}.$$

By setting $\delta < 9\varepsilon/(1 + 6\varepsilon)$, we satisfy the required inequality and achieve the inapproximability gap in the theorem statement. $\square$

### D.3 Proof of Lemma 4.4

Let $G = (V, E)$ be a simple undirected graph. Our goal is to embed the vertices of $V$ into $\mathbb{R}^d$, for some $d \geq 1$, in a way that encodes the adjacency structure of $G$. Concretely, we want to construct a function $h_G : V \to \mathbb{R}^d$ such that:

- $\|h_G(u) - h_G(v)\|_2 = 1$ if $\{u, v\} \notin E$, and
- $\|h_G(u) - h_G(v)\|_2 \leq 1 - \varepsilon_G$ if $\{u, v\} \in E$,

for the largest possible value of $\varepsilon_G \in (0, 1]$.

**Construction.** Let $n = |V|$ and $m = |E|$. Augment $G$ by adding a self-loop to each node to get $G' = (V, E')$. We embed $V$ using $G'$ since each node now has positive degree. Let $\deg'(v)$ be the degree of $v$ in $G'$ and $N'(v)$ be the neighborhood of $v$ in $G'$.

Define a total ordering on $E'$ (e.g., lexicographically by sorted endpoints $\{u, v\}$). Each edge $e \in E'$ corresponds to an index in the embedding dimension $d := |E'| = m + n$. We consider the embedding function that acts as a degree-normalized adjacency vector:

$$h_G(v)_e = \begin{cases} \sqrt{\frac{1}{2\deg'(v)}} & \text{if } v \in e, \\ 0 & \text{if } v \notin e. \end{cases} \tag{12}$$

**Lemma 4.4.** *Let $G = (V, E)$ be a simple undirected graph with $n = |V|$, $m = |E|$, and max degree $\Delta$. There exists an embedding $h_G : V \to \mathbb{R}^{n+m}$ such that if $\{u, v\} \in E$ then*

$$\|h_G(u) - h_G(v)\|_2 \leq 1 - \frac{1}{2(\Delta + 1)},$$

*and if $\{u, v\} \notin E$ then $\|h_G(u) - h_G(v)\|_2 = 1$.*

*Proof.* If $\{u,v\} \notin E$, then we have

$$\|h_G(u) - h_G(v)\|_2^2 = \sum_{e \in N'(u)} \left(\sqrt{\frac{1}{2\deg'(u)}} - 0\right)^2 + \sum_{e \in N'(v)} \left(\sqrt{\frac{1}{2\deg'(v)}} - 0\right)^2$$

$$= \left(\frac{1}{2} \sum_{e \in N'(u)} \frac{1}{\deg'(u)}\right) + \left(\frac{1}{2} \sum_{e \in N'(v)} \frac{1}{\deg'(v)}\right)$$

$$= \frac{1}{2} + \frac{1}{2}$$

$$= 1.$$

This follows because the only index where both embeddings can be nonzero is $\{u,v\}$, if it exists. Now suppose that $\{u,v\} \in E$. It follows that

$$\|h_G(u) - h_G(v)\|_2^2$$

$$= \sum_{e \in N'(u)\setminus\{v\}} \frac{1}{2\deg'(u)} + \sum_{e \in N'(v)\setminus\{u\}} \frac{1}{2\deg'(v)} + \left(\sqrt{\frac{1}{2\deg'(u)}} - \sqrt{\frac{1}{2\deg'(v)}}\right)^2$$

$$= \frac{\deg'(u)-1}{2\deg'(u)} + \frac{\deg'(v)-1}{2\deg'(v)} + \left(\frac{1}{2\deg'(u)} + \frac{1}{2\deg'(v)} - 2\sqrt{\frac{1}{4\deg'(u)\deg'(v)}}\right)$$

$$= \frac{1}{2} + \frac{1}{2} - \sqrt{\frac{1}{\deg'(u)\deg'(v)}}$$

$$\leq 1 - \frac{1}{\Delta+1}.$$

The previous inequality follows from $\deg'(v) = \deg(v) + 1 \leq \Delta + 1$. For any $x \in [0,1]$, we have

$$\sqrt{1-x} \leq 1 - \frac{x}{2},$$

so it follows that

$$\|h_G(u) - h_G(v)\|_2 \leq \sqrt{1 - \frac{1}{\Delta+1}} \leq 1 - \frac{1}{2(\Delta+1)},$$

which completes the proof. $\qquad\square$

### D.4 Proof of Theorem 4.5

**Theorem 4.5.** MDMS *is* APX*-complete for the Euclidean metric if $g$ is a linear function.*

*Proof.* We build on the hardness of approximation for the maximum independent set problem for graphs with maximum degree $\Delta = 3$. Alimonti and Kann [4, Theorem 3.2] showed that this problem is APX-complete, so there exists an $\varepsilon_0 > 0$ such that there is no polynomial-time $(1 - \varepsilon_0)$-approximation algorithm unless NP = P. Hence, there exists a graph $G$ with max degree $\Delta = 3$ and a threshold integer value $k$ such that it is NP-hard to distinguish between the following two cases:

- YES instance: graph $G$ has an independent set of size $k$.

- NO instance: graph $G$ does not have an independent set of size greater than $(1 - \varepsilon_0)k$.

We reduce this instance of bounded-degree maximum independent set to MDMS with objective function (2) as follows. Embed each node of the graph $G$ into Euclidean space using the function $h_G(v)$ in Lemma 4.4. We use the same threshold value of $k$ (between YES and NO instances above) for the cardinality constraint on set $S$, and we set each weight $w(v) = \alpha/k$ for some parameter $\alpha$ that we set later in the proof. We also set $\lambda = 1 - \alpha$.

In a YES instance, selecting an independent set of size $k$ as the set $S$ results in the maximum value of objective (2):

$$\text{OPT} = \alpha \cdot \frac{1}{k} \cdot k + (1 - \alpha) \cdot 1 = 1,$$

since $\|h_G(u) - h_G(v)\|_2 = 1$ for any two distinct points $u, v \in S$ since there is no edge between $u$ and $v$ in graph $G$.

In a NO instance, the best objective value that can be achieved in polynomial-time is the maximum of the following two scenarios: (a) selecting $k$ points with minimum distance at most $1 - 1/(2(\Delta + 1)) = 1 - 1/8$, or (b) selecting at most $(1 - \varepsilon_0)k$ vertices forming an independent set with minimum distance equal to $1$. The maximum value obtained by any polynomial-time algorithm is then

$$\begin{aligned}
\text{ALG} &= \max\{\alpha(1 - \varepsilon_0) + (1 - \alpha) \cdot 1, \alpha + (1 - \alpha)(1 - 1/8)\} \\
&= \max\{1 - \varepsilon_0 \cdot \alpha, (7 + \alpha)/8\}.
\end{aligned}$$

We make these two terms equal by setting $\alpha = 1/(1 + 8\varepsilon_0)$. Therefore, the gap between the maximum value any algorithm can achieve in the NO case and the optimum value in the YES case is upper bounded by

$$1 - \varepsilon_0 \cdot \alpha = 1 - \frac{\varepsilon_0}{1/\varepsilon_0 + 8} = 1 - \varepsilon_1.$$

Since $\varepsilon_0 > 0$ is a constant, $\varepsilon_1 := \varepsilon_0/(1/\varepsilon_0 + 8) > 0$ is also a constant. This completes the proof of APX-completeness. $\square$

# E  Additional details for Section 5

## E.1  Synthetic dataset

We extend our comparison of baseline algorithms in Section 5.1 by sweeping over values of $\alpha, \beta$ in the objective function.

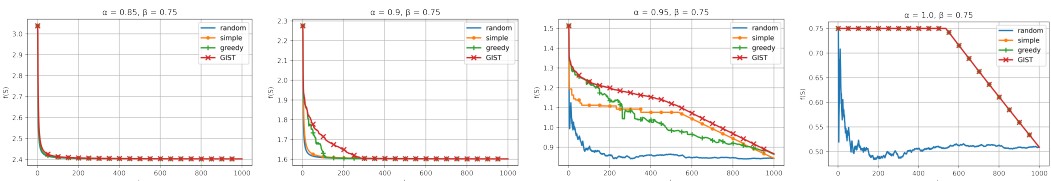

*Figure 2:* Baseline comparison with $\alpha \in (0.85, 0.90, 0.95, 1.00)$ and $\beta = 0.75$.

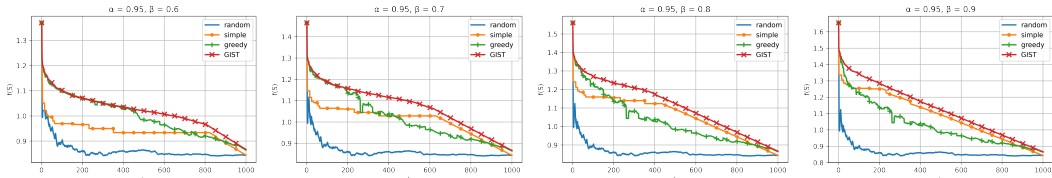

*Figure 3:* Baseline comparison with $\alpha = 0.95$ and $\beta \in (0.60, 0.70, 0.80, 0.90)$.

## E.2  Image classification

**Hyperparameters for ImageNet classification.**  We generate predictions and embeddings for all points using a coarsely-trained ResNet-56 model [29] trained on a random 10% subset of ImageNet [54]. We use SGD with Nesterov momentum 0.9 with 450/90 epochs. The base learning rate is 0.1, and is reduced by a tenth at 5, 30, 69, and 80. We extract the penultimate layer features to produce 2048-dimensional embeddings of each image. We use the same hyperparameters as the original ResNet paper [29] with budgets and one-shot subset selection experiments designed in the same manner as [49].

**Running times.** The end-to-end running time is dominated by ImageNet model training, which takes more than a few hours even with accelerators (e.g., GPU/TPU chips). The subset selection algorithms that use margin and submodular sampling range between 3–4 minutes per run on an average. GIST subset selection is similar to the margin or submodular algorithms for a given distance threshold $d$. By using parallelism for different $d$ values, we can keep the GIST-submod subset selection runtime the same as the submod algorithm. In summary, the actual subset selection algorithm step is extremely fast (nearly negligible) compared to the ImageNet training time. Furthermore, we can exploit distributed submodular subset selection algorithms that can even handle billions of data points efficiently [15].

