# OpenReview forum: "GIST: Greedy Independent Set Thresholding for Max-Min Diversification with Submodular Utility"
_NeurIPS.cc/2025/Conference — NeurIPS 2025 poster_

### Official Review · Reviewer_i9Xd · 2025-06-21

**Clarity:** 3
**Significance:** 3
**Originality:** 3
**Rating:** 5
**Confidence:** 4

**Summary:**

The paper studies the min-distance variant of the submodular+diversification problem with a cardinality constraint. An iterative greedy algorithm (GIST) is proposed, and its approximation guarantees are analysed. The results are the paired with approximation hardness bounds. Experimental results indicates the utility of GIST in objective values and downstream performance in one-shot image classifcation (w. GIST acting as a sample selector).

**Questions:**

Line 591: do you mean "the distance is 2 if there is NO edge"?

**Ethical Concerns:**

["NO or VERY MINOR ethics concerns only"]

**Final Justification:**

The rebuttals sufficiently addressed the comments, the most concerning of which refers to the novelty of the proposed method.

**Limitations:**

yes

**Quality:**

3

**Strengths And Weaknesses:**

The contribution is significant as it gives a practical method for a problem relevant to the field of machine learning. As far as I know, this variant of the problem is studied for the first time in this paper.

The paper is mostly well-written. The contributions are clearly stated and met. The theoretical results are strong and comprehensive. The proofs are made easy to follow with the use of intuitive connection to graph-theoretical results. The findings give a narrow bound for the approximation factor.

However, the baseline greedy heuristic is not examined (i.e, choosing $\text{argmax}_{v}f(x|T)$). If there is a reason for this (e.g., no constant-factor approximation), it should be stated.

Some comments regarding Section 5. It is not stated which value of the error term ($\varepsilon$) is used for GIST in the experiments. Also, the results in Figure 1 would be more meaningful if normalized by f(OPT). Although, considering that these are random instances, the lack of best-known values is excusable. Finally, for the image classification task, statistical tests are called for since randomness is involved. The test procedure could be explained in Appendix.

While the results on linear utility is interesting academically, this special case is not really relevant to the suggested applications of the problem (e.g., data sampling). Different directions that might be more relevant include specific distance metrics, or other types of constraints (an example of fairness constraints is mentioned).

---

> ### Author Rebuttal · Authors · 2025-07-30
>
> Thank you for your supportive review. Please see our answers to your questions below.
>
> > … the baseline greedy heuristic is not examined … If there is a reason for this (e.g., no constant-factor approximation), it should be stated.
>
> The objective function $f$ is highly non-monotone since the diversity term can decrease as we add more items. Consequently, the standard `greedy` algorithm, which rejects any item with a negative marginal gain, can have an arbitrarily poor performance.
>
> We demonstrate this with a hard instance. Let the submodular part of the objective be $g(S) = |S|$. For the diversity component, define the distance between a specific pair $(u,v)$ to be $\text{dist}(u, v) = 2 + 2 \varepsilon$, while for all other pairs $(x,y) \ne (u,v)$, we have $\text{dist}(x, y) = 1 + \varepsilon$. The `greedy` algorithm first selects set $\{u, v\}$, achieving a value of $f(\{u,v\}) = g(\{u,v\}) + \text{dist}(u,v) = 4+2\varepsilon$. The algorithm then terminates because adding any subsequent item gives a submodular gain of 1 but causes a diversity loss of $1 + \varepsilon$, resulting in a negative marginal gain.
>
> An optimal solution of size $k$, however, can achieve a value of at least $k$ from the submodular term alone. The resulting approximation ratio for `greedy` is therefore at most $(4 + 2\varepsilon) / k$​, which approaches 0 as $k$ grows. Thus, `greedy` offers no constant-factor approximation guarantee. Moreover, modifying `greedy` to accept items with negative marginal value does not solve this, as one can construct instances where an optimal solution's value is dominated by a diversity term that any $k$-sized set diminishes to zero. We will add this discussion about `greedy` to the paper for completeness.
>
> > While the results on linear utility is interesting academically, this special case is not really relevant to the suggested applications of the problem (e.g., data sampling).
>
> We agree that using a linear utility could be too special in some cases. That said, it helps understand the theoretical landscape of the optimization problem and set of tractable objectives.
>
> > It is not stated which value of the error term ($\varepsilon$) is used for GIST in the experiments.
>
> This is an oversight on our end. Thank you for catching this.
>
> In the warm-up synthetic experiments (Section 5.1), we set $D$ to be all possible pairwise distances since $n = 1000$. The running time is $O(nk \cdot n^2)$ and the algorithm yields an exact $2/3$-approximation ratio.
>
> In the image classification experiments (Section 5.2), we used $\varepsilon = 0.05$. We will add these details to the paper.
>
> > Finally, for the image classification task, statistical tests are called for since randomness is involved. The test procedure could be explained in Appendix.
>
> We followed the experimental setup in “Active learning for convolutional neural networks: A core-set approach” [Sener–Savarese, ICLR 2017] by reporting the mean and standard deviation over multiple trials to be backwards compatible with the literature. We will include the raw loss values from each trial in the appendix to increase transparency.
>
> > Line 591: do you mean "the distance is 2 if there is NO edge"?
>
> No, "the distance between a pair of points is 2 if there is an edge" is the correct formulation because the YES instance in the hardness example has a clique of size $k$, e.g., $k$ vertices forming a complete subgraph and objective value in Eq. (11). Alternatively, one can phrase the NP-hard instance with independent sets instead of cliques; in this case, we should set distance to 2 when there is no edge.

---

> > ### Comment · Reviewer_i9Xd · 2025-08-05
> >
> > Thanks for answering the question. I will keep my score as I assumed the raised issues would be addressed anyways.

---

### Official Review · Reviewer_tLCc · 2025-06-26

**Clarity:** 3
**Significance:** 3
**Originality:** 3
**Rating:** 5
**Confidence:** 3

**Summary:**

The problem considered in this paper is to maximize a set function $f(S) = g(S) + \lambda div(S)$ subject to $|S| \le k$, where $g$ is a non-negative monotone submodular function and $div(S)$ is the minimum distance between two points in $S$. In data subset selection problems, $g(S)$ represents coverage of information of $S$, and $div(S)$ represents diversity of $S$. This paper proposes a clean $(0.5-\epsilon)$-approximation algorithm with running time $O(n k \epsilon^{-1} \log(1/\epsilon))$, while also providing an impossibility result of $0.5584$-approximation under P $\neq$ NP assumption. In the special case of linear $g$, the proposed algorithm is $(2/3-\epsilon)$-approximation and the impossibility result is tight. Another hardness result is the APX-completeness in the case of linear $g$ and Euclidean metric. The experimental result on data subset selection from ImageNet shows that the proposed method outperforms standard benchmarks (such as random selection and choosing data close to the margin) in terms of prediction accuracy.

**Questions:**

No question.

**Ethical Concerns:**

["NO or VERY MINOR ethics concerns only"]

**Final Justification:**

There is no specific change in my evaluation, and I keep my positive score.

**Limitations:**

Yes

**Quality:**

3

**Strengths And Weaknesses:**

Strengths:
- The algorithmic idea is simple and can be applicable to many other problems in the future. An important observation is that monotone submodular maximization with an independent set constraint has a bicriteria $1/2$-approximation algorithm. So if $div(S^*)$ for the optimal set $S^*$ is known in advance, $1/2$-approximation can be obtained. Since it is unknown, the proposed algorithm applies the bicriateria algorithm for multiple possible values of $div(S^*)$ and takes the maximal one.
- It is nice to see the tight $(2/3+\epsilon)$-approximation hardness result for linear $g$.
- The experimental results are also very good. They show superiority of the proposed method not only in terms of the objective function, but also in terms of the prediction accuracy in data subset selection problems.

---
Weaknesses:
- I have no concern.


---
Typos:
- In the pseudocode Algorithm 1, the domain of $g$ should be $2^V$, not $V$.

---

> ### Author Rebuttal · Authors · 2025-07-30
>
> Thank you for the supportive review and for catching the typo about the domain of $g$. We appreciate the careful read.

---

### Official Review · Reviewer_syZ1 · 2025-07-01

**Clarity:** 4
**Significance:** 3
**Originality:** 3
**Rating:** 5
**Confidence:** 4

**Summary:**

This paper studies a novel subset selection problem named max-min diversification with monotone submodular utility, which can be applied in feature selection, recommendation systems, and data summarization. First, this paper formalizes the problem and presents a simple warm-up algorithm. Next, this paper presents the GIST algorithm and proves that a 1/2-approximation guarantee can be achieved. In the special case of linear utility functions, the approximation guarantee can be further improved to 2/3. This paper then demonstrates the hardness of this problem under various restrictions. Finally, this paper presents two experiments, one on synthetic data and the other on real-world data, to demonstrate its performance in various tasks, including data sampling.

**Questions:**

See weaknesses.

**Ethical Concerns:**

["NO or VERY MINOR ethics concerns only"]

**Final Justification:**

Thanks for the rebuttal. I will keep my score.

**Limitations:**

Yes.

**Paper Formatting Concerns:**

No paper formatting concerns.

**Quality:**

3

**Strengths And Weaknesses:**

**Strengths**

S1. This is a comprehensive piece of work, comprising a warm-up algorithm, a formal algorithm, a substantial proof of approximation guarantee and harness, and experiments that demonstrate the algorithm's strengths. The lemmas and theorems in this paper are well-supported by theoretical proofs, and the algorithm is well-supported by sufficient experimental results.

S2. The submission is clearly written and well organized. The proof process is clear enough for readers to understand its correctness. The experimental setting is described well, allowing readers to understand and potentially reproduce the results.

S3. The topic and results are impactful for the community and society, as they address the diversity of data selection, and the algorithm can be implemented in many real-world applications with minimal expenses.

**Weaknesses**

W1. The analysis for the time complexity can be complemented in detail. For instance, Theorem 3.1 presents the time complexity of GIST, but the proof or demonstration is not included.

W2. The experiments included in this paper are convincing enough, but comparing the running time between GIST and other traditional algorithms can further indicate the performance of the GIST algorithm.

---

> ### Author Rebuttal · Authors · 2025-07-30
>
> Thank you for your supportive review and articulating the strengths of our paper. Please see our answers to your questions below.
>
> > The analysis for the time complexity can be complemented in detail. For instance, Theorem 3.1 presents the time complexity of GIST, but the proof or demonstration is not included.
>
> Thanks for catching the missing oracle query complexity analysis in the proof of Theorem 3.1. We will include it in the final version. It is essentially the number of distance thresholds that the algorithm tries times the query complexity of the `GreedyIndependentSet` subroutine.
>
> > The experiments included in this paper are convincing enough, but comparing the running time between GIST and other traditional algorithms can further indicate the performance of the GIST algorithm.
>
> The end-to-end running time is dominated by training ImageNet models, which takes more than a few hours even with several accelerators (e.g., GPU/TPU chips). The subset selection algorithms that use margin and submodular sampling range between 3–4 minutes per run on an average. GIST subset selection is similar to the margin or submodular algorithms for a given distance threshold $d$. By using parallelism for different $d$ values, we are able to keep the `GIST-submod` subset selection runtime the same as the `submod` algorithm. In summary, the actual subset selection algorithm step is extremely fast (nearly negligible) compared to the ImageNet training time. It also runs on cheaper CPUs compared to ML accelerators. We will be happy to add these details in the final version.

---

> > ### Comment · Reviewer_syZ1 · 2025-08-05
> >
> > Thanks for the rebuttal. I will keep my score.

---

### Official Review · Reviewer_t8m7 · 2025-07-09

**Clarity:** 3
**Significance:** 2
**Originality:** 2
**Rating:** 4
**Confidence:** 3

**Summary:**

This paper proposes and develops algorithms for an optimization problem where we maximize a monotone submodular function plus a weighted diversity function. This formulation should capture applications such as data summarization where we not only seek coverage (with the monotone submodular function), but also diversity (through the weighted diversity function). In particular, they consider a max-min diversification term for the second part of the funciton. This objective is not submodular, so novel algorithms must be proposed. While related objectives have been studied several times (e.g. [13]), their particular objective has not been.

They develop the algorithm GIST, which is a 1/2 approximation algorithm. GIST is simple, in fact it looks like they are making guesses of the maxmin distance and running greedy algorithms for each of those guesses. In addition, they develop hardness of approximation results for their novel problem, and several other results for the special case of linear submodular functions. Finally, an experimental section is included where alternative baselines are compared to GIST.

**Questions:**

- See weaknesses above

**Ethical Concerns:**

["NO or VERY MINOR ethics concerns only"]

**Final Justification:**

I have read the rebuttal of the authors, as well as the other reviews. I maintain my opinion that the paper should be accepted.

**Limitations:**

yes

**Quality:**

3

**Strengths And Weaknesses:**

Strengths
- They propose a novel optimization problem which seems to have applications.
- Novel algorithms are developed with constant factor approximation guarantees.
- They have a hardness result for their problem.
- They have an experimental section to back up their theoretical claims.

Weaknesses
- I am unclear about how this new optimization problem is beneficial over existing formulations that capture both coverage and diversity. In particular, within non-monotone submodular optimization, people have considered objectives such as Determinantal Point Processes (DPP) [1] in order to get diverse sets. If we were to take a monotone submodular function and add a weighted functions such as a DPP, then overall we would have a non-monotone submodular function that captures both coverage and diversity, and there exists many algorithms for this setting. What is the benefit to applications of the formulation in this paper over this alternative one?
- The development and analysis of GIST looks like it doesn't require a ton of theoretical novelty, it sort of reminds me of a combination of the greedy algorithm, the streaming algorithms in submodular optimization, and the doubling algorithm. But it appears new for this problem.

[1]Kulesza, Alex, and Ben Taskar. "Determinantal point processes for machine learning." Foundations and Trends® in Machine Learning 5.2–3 (2012): 123-286.

---

> ### Author Rebuttal · Authors · 2025-07-30
>
> Thank you for your supportive review and careful observations. Please see our answers to your questions below.
>
> > What is the benefit to applications of the formulation in this paper over a [non-monotone submodular DPP-based] alternative?
>
> We agree that determinantal point processes (DPPs) are an effective method for modeling diversity, as they operate as log-submodular, non-monotone functions that capture the full range of interactions between selected items.
>
> Our work explores a valid alternative. We model diversity primarily through pairwise interactions, while higher-order effects are captured implicitly through the submodular term from the theoretical perspective. For our target applications, we observed that this pairwise approach was sufficient to achieve the desired level of diversity. We also note that with the scale of datasets these days, capturing higher-order interactions between points (beyond pairwise) is very challenging and makes the computational cost prohibitive unless very efficient algorithms are designed for the specific objectives, e.g., log-DPP plus submodular.
>
> A key distinction between these methods also lies in their theoretical approximation guarantees. Using a log-DPP term for diversity would render the objective non-monotone submodular (as you observed), restricting the best possible approximation guarantee to be at most $0.478$ subject to a cardinality constraint $k$ [1], unless a better approximation algorithm is designed specifically for this DPP-based objective. In contrast, `GIST` gives a $1/2$-approximation guarantee and designing an algorithm with a better than $1/2$-approximation for MDMS remains an interesting open question.
>
> > The development and analysis of GIST looks like it doesn't require a ton of theoretical novelty … but it appears new for this problem.
>
> We believe the simplicity of GIST is an advantage, making it a more practical solution and accessible in a wider range of applications. We argue that the two-stage structure of the algorithm by "guessing" the target diversity (distance threshold) and optimizing based on that initial decision is crucial. We provide the following evidence to show that the standard `greedy` approach *fails to achieve any constant-factor approximation guarantee*.
> Our objective function $f$ is highly non-monotone since the diversity term can decrease as we add more items. Consequently, the standard `greedy` algorithm that adds the best item at each step while rejecting any item with a negative marginal gain, can have an arbitrarily poor performance.
>
> We demonstrate this with a hard instance. Let the submodular part of the objective be $g(S) = |S|$. For the diversity component, define the distance between a specific pair $(u,v)$ to be $\text{dist}(u, v) = 2 + 2 \varepsilon$, while for all other pairs $(x,y) \ne (u,v)$, we have $\text{dist}(x, y) = 1 + \varepsilon$. The `greedy` algorithm first selects set $\{u, v\}$, achieving a value of $f(\{u,v\}) = g(\{u,v\}) + \text{dist}(u,v) = 4+2\varepsilon$. The algorithm then terminates because adding any subsequent item gives a submodular gain of 1 but causes a diversity loss of $1 + \varepsilon$, resulting in a negative marginal gain.
>
> An optimal solution of size $k$, however, can achieve a value of at least $k$ from the submodular term alone. The resulting approximation ratio for `greedy` is therefore at most $(4 + 2\varepsilon) / k$​, which approaches 0 as $k$ grows. Thus, `greedy` offers no constant-factor approximation guarantee.
>
> **References**
>
> [1] B. Qi. “On maximizing sums of non-monotone submodular and linear functions.” Algorithmica, 2024.

---

### Decision · Program_Chairs · 2025-09-17

**Decision:**

Accept (poster)

**Comment:**

The paper introduces and studies a novel problem formulation for size-constrained subset selection where the goal is to maximize the sum of a monotone submodular function and a diversity function that captures the minimum distance between any two points in the chosen subset. The paper gives an algorithm for the problem that achieves a 1/2 approximation and it shows that the problem is NP-hard to approximate within a factor of 0.5584. The paper shows an improved approximation guarantee when the submodular function is linear and it provides an experimental evaluation to support the theoretical results. The reviewers appreciated the main contributions of the paper, including the novel problem formulation that provides a more tractable formulation that incorporates diversity, the strong theoretical guarantees established, and the experimental evaluation. The author response and the subsequent discussion sufficiently addressed the reviewers' questions and concerns. Overall, there was consensus among the reviewers that the paper meets the threshold for acceptance.